# Glycolytic Inhibitors Potentiated the Activity of Paclitaxel and Their Nanoencapsulation Increased Their Delivery in a Lung Cancer Model

**DOI:** 10.3390/pharmaceutics14102021

**Published:** 2022-09-23

**Authors:** Andrea Cunha, Ana Catarina Rocha, Flávia Barbosa, Ana Baião, Patrícia Silva, Bruno Sarmento, Odília Queirós

**Affiliations:** 1UNIPRO—Oral Pathology and Rehabilitation Research Unit, University Institute of Health Sciences (IUCS), CESPU, 4585-116 Gandra, Portugal; 2DCM—Departamento de Ciências Médicas, Universidade de Aveiro, 3810-193 Aveiro, Portugal; 3i3S—Instituto de Investigação e Inovação em Saúde, Universidade do Porto, 4200-135 Porto, Portugal; 4INEB—Instituto de Engenharia Biomédica, Universidade do Porto, 4200-135 Porto, Portugal; 5ICBAS—Instituto de Ciências Biomédicas Abel Salazar, Universidade do Porto, 4050-313 Porto, Portugal; 6TOXRUN—Toxicology Research Unit, University Institute of Health Sciences (IUCS), CESPU, 3810-193 Gandra, Portugal

**Keywords:** tumor metabolism, Warburg effect, 3-bromopyruvate, dichloroacetate, 2-deoxyglucose, nanoparticles

## Abstract

Antiglycolytic agents inhibit cell metabolism and modify the tumor’s microenvironment, affecting chemotherapy resistance mechanisms. In this work, we studied the effect of the glycolytic inhibitors 3-bromopyruvate (3BP), dichloroacetate (DCA) and 2-deoxyglucose (2DG) on cancer cell properties and on the multidrug resistance phenotype, using lung cancer cells as a model. All compounds led to the loss of cell viability, with different effects on the cell metabolism, migration and proliferation, depending on the drug and cell line assayed. DCA was the most promising compound, presenting the highest inhibitory effect on cell metabolism and proliferation. DCA treatment led to decreased glucose consumption and ATP and lactate production in both A549 and NCI-H460 cell lines. Furthermore, the DCA pretreatment sensitized the cancer cells to Paclitaxel (PTX), a conventional chemotherapeutic drug, with a 2.7-fold and a 10-fold decrease in PTX IC_50_ values in A549 and NCI-H460 cell lines, respectively. To increase the intracellular concentration of DCA, thereby potentiating its effect, DCA-loaded poly(lactic-*co*-glycolic acid) nanoparticles were produced. At higher DCA concentrations, encapsulation was found to increase its toxicity. These results may help find a new treatment strategy through combined therapy, which could open doors to new treatment approaches.

## 1. Introduction

Non-small cell lung cancer is one of the most common malignant tumors in men and women and a leading cause of mortality worldwide [1,2]. The high metabolic rate, characteristic of tumor cells, depends on several factors, from tumor glycolysis has been considered one of the most important [3]. One of the hallmarks of cancer is its altered metabolism, which includes a metabolic shift in energy production, from oxidative phosphorylation (OXPHOS) to glycolysis even under normoxia, which is known as the Warburg effect or aerobic glycolysis [4]. Although OXPHOS is more efficient in generating ATP, glycolysis meets the high and rapid demand of energy by tumor cells through the upregulation of the glycolytic flux, metabolizing glucose at high rates with increased lactate production [5]. Augmented consumption of extracellular glucose is achieved through the overexpression of glucose transporters (GLUTs), whereas lactate export is ensured by a proton symport mechanism, mediated by the monocarboxylate transporters (MCTs), in particular by MCT1 and MCT4, present at elevated levels in the plasma membrane of tumor cells [3,6,7]. MCT1/4 are closely associated with CD147, a chaperone required for their activity [8,9]. Silencing MCT1 or MCT4, in combination with CD147 by siRNA or their inhibition by small-molecule inhibitors, induced a significant reduction in glycolytic flux and cell proliferation [6,10,11]. Lactate efflux allows a self-defense strategy of tumor cells through the simultaneous export of protons, enabling the maintenance of a normal intracellular pH and, thus, avoiding apoptosis [7]. Additionally, the acidic extracellular environment, created by lactate and proton efflux, is associated with tumor aggressiveness, namely the suppression of the immune system and the multiple drug resistance (MDR) phenotype [3,7,8].

MDR is a mechanism of resistance developed by several cancer cells to multiple chemotherapeutic drugs, often with different structures and targets, representing one of the main obstacles to treatment success [3,7,12]. Many studies have shown an association between MDR and the Warburg effect, suggesting that the high glycolytic rate promotes tumor cell resistance to antitumor treatment. High glycolytic rates are associated with increased lactate secretion and extracellular space acidification, leading to poorer drug stability and, consequently, lower drug efficacy [3]. In this way, the metabolic differences between normal and tumor cells offer new opportunities for developing powerful strategies for cancer therapies. Glycolytic inhibitors (GIs), such as 2-deoxyglucose (2DG)—C_6_H_12_O_5_, dichloroacetate (DCA)—C_2_HCl_2_O_2_—and 3-bromopyruvate (3BP)—C_3_H_3_BrO_3_—have been assayed as putative antitumor agents that target glycolysis, hampering this metabolic pathway or, in the case of DCA, redirecting pyruvate to acetyl-CoA synthesis [13,14].

Similar to glucose, 2DG uses GLUT transporters to enter the cell, where it competes with glucose in the first step of its intracellular metabolism, phosphorylated by hexokinase II (HKII). 2DG is converted into deoxyglucose-6-phosphate, which is not further metabolized, blocking glycolysis [15,16,17]. 2DG was shown to mediate cancer cell death in normoxic cancer cells due to unspecific glycosylation of proteins. Furthermore, combined treatment with 2DG enhanced the efficacy of conventional anticancer drugs such as paclitaxel (PTX) in osteosarcoma, non-small cell lung cancer xenografts and Ehrlich hepatoma-bearing mice [18,19,20]. However, further investigation is necessary to understand the main molecular mechanisms underlying the therapeutic efficacy of 2DG [16,21].

3BP, a structural analog of pyruvate, is a potent alkylating agent whose effect is verified by inhibiting tumor cell metabolism and promoting cellular ATP depletion [8]. One of the significant targets of 3BP is the glycolytic enzyme HKII, although it can also inhibit mitochondrial metabolism [8,22]. Various data showed that 3BP exhibits high anticancer activity, e.g., in advanced stage hepatomas and human prostate cancer, also considered a potent MDR reversal modulator [22,23]. However, some studies reported an association between its administration and some cases of cancer patient deaths [24]. Thus, other efforts are needed to clarify this compound’s toxicity and side effects and determine the correct dose to administer.

Lastly, DCA, a known activator of Pyruvate Dehydrogenase (PDH) through its inhibition of PDH kinase (PDK), is a small molecule that can reverse the Warburg effect since it redirects pyruvate from the glycolytic flux to its oxidative metabolism [25,26]. The stimulation of oxidative metabolism by DCA leads to the production of reactive oxygen species (ROS), which plays an important role in the induction of apoptosis [27,28]. Furthermore, DCA was observed to increase the activity of p53, which also contributes to tumor cell apoptosis and, consequently, to the decrease of cell proliferation [28]. Given its promising features, DCA is currently being evaluated in clinical trials in patients with solid cancers (NCT01029925, NCT0056410, NCT01111097) [29,30,31,32]. In addition, preclinical results indicate that DCA may synergize well with chemotherapeutic agents such as 5- fluorouracil and cisplatin [30,33].

GIs may be associated with chemotherapeutic drugs to overcome the MDR phenotype and open a new door for cancer therapies [7]. Here, we aimed to study cancer metabolism after treatment with GIs and their influence on combined treatment with chemotherapeutic drugs. For this purpose, we have chosen PTX as a chemotherapeutic drug since it is a conventional drug used as first-line chemotherapy for the majority of types of cancer, namely lung cancer [34]. However, it induces a large range of side effects, for which it is desirable to use lower but effective concentrations. Lung cancer has become one of the most frequently diagnosed cancers in recent years, representing the leading cause of cancer deaths in men and the second cause in women [35]. The recommended treatment for patients with advanced lung cancer involves systemic platinum-based chemotherapy (e.g., cisplatin) combined with taxanes (such as PTX) [36,37]. PTX is an antimitotic drug that induces tumor growth inhibition [38]. However, it is often associated with chemotherapy failure due to increased acquisition of resistance by tumor cells. Such resistance has been assigned to various mechanisms, among which P-Glycoprotein (Pgp) overexpression is one of the most important [12,38,39]. Pgp has been described as the main cause of MDR phenomena in several types of cancer. The combined use of GIs and PTX could be a way to overcome this problem since it is known that cells expressing MDR proteins, such as Pgp, require ATP as the energy source to pump out the drug substrates [40,41].

Chemotherapy side effects are due to the lack of specificity of conventional anticancer drugs that target common cell processes, which may compromise treatment success. Thus, multiple nanoformulations have been developed to avoid such effects since they can increase the accumulation of drugs at tumor sites [40,42]. In this way, to improve the intracellular delivery of GIs and regarding an efficient targeting of cancer cells, we assayed the encapsulation of GIs into polymeric nanoparticles (NPs). Among polymeric NPs, self-assembled monolayers, methoxy poly(ethylene glycol)-*b*-poly(allyl glycidyl ether)-*b*-poly(e-caprolactone), nanofibers and poly(lactic-*co*-glycolic acid) (PLGA) have been widely used due to their properties, as they are biocompatible and biodegradable polymers, FDA-approved and allow a good intracellular delivery of drugs [43,44,45].

The main goal of this study was to open doors for new therapeutic strategies using GIs to overcome the resistance to conventional drugs and verify if the use of NPs could improve the delivery of these inhibitors to tumor cells. To achieve this goal, we studied the effect of 3BP, DCA and 2DG in small-cell lung cancer cells (used as a tumor model) and their efficacy when combined with chemotherapeutic drugs, namely PTX. We also designed a strategy to improve the intracellular accumulation of GIs in cancer cells through drug delivery nanosystems.

## 2. Materials and Methods

### 2.1. Cell Culture

NCI-H460 and A549 were used as lung cancer cell line models and HPAEpic as a normal lung cell line model. All cell lines were obtained from the American Type Culture Collection (ATCC) and grown as monolayers in a humidified incubator with 5% of CO_2_ at 37 °C. Before each assay, after seeding, cells were incubated overnight, allowing them to stabilize and adhere, before exposure to drugs.

NCI-H460 cells were subcultured in Roswell Park Memorial Institute medium 1640 (RPMI-1640, Lonza, Basel, Switzerland), supplemented with 10% of heat-inactivated fetal bovine serum (FBS, Biochrom, Cambridge, UK) and 1% of penicillin/streptomycin antibiotics (Lonza). A549 and HPAEpic were subcultured in Dulbecco’s Modified Eagle Medium (DMEM, Lonza), supplemented with 10% of FBS, 1% Non-Essential Amino Acids (NEAA, Sigma-Aldrich, St. Louis, MO, USA) and 1% of penicillin/streptomycin antibiotics (Lonza).

For all the assays performed in 96-well plates, the plates were seeded with 200 μL of cell suspension, corresponding to 10,000 cells/well for NCI-H460 cells, 15,000 cells/well for A549 and 25,000 cells/well for HPAEpic. In 6-well plate assays, 1.5 mL of cell suspension were used, corresponding to 2.4 × 10^5^ cells/well for A549, 1.6 × 10^5^ cells/well for NCI-H460 and 4.0 × 10^5^ cells/well for HPAEpic cells.

### 2.2. Drugs

A commercial solution of PTX was purchased from Hospira, Portugal. The GIs 3BP, 2DG and DCA (Sigma-Aldrich) were dissolved in fresh cold PBS to prepare 10, 300 and 1000 mM stock solutions, respectively, from which the working solutions were prepared by dilution. All stock solutions were filtered and used immediately.

### 2.3. Cell Viability Assays

Cell viability was assessed by the sulforhodamine B (SRB) assay, as previously described [8]. To this purpose, cells in the exponential growth phase were seeded in 96-well plates and treated with 3BP or DCA for 24 h and with 2DG or PTX during 48h. Untreated cells were used as controls, with the drug volume replaced by the same amount of the respective vehicle, being considered as 100% of viability. After treatment, adherent cells were fixed at 4 °C, for 1 h with 25 μL of 50% (*w/v*) TCA. The plates were then washed with water, air-dried and stained with 50 µL of 0.4% (*w/v*) SRB in 1% (*v/v*) acetic acid for 30 min. After staining, the plates were rinsed with 1% acetic acid and air-dried. The SRB incorporated was solubilized with 100 µL of 10 mM Tris buffer, and the absorbance of each well was measured at 515 nm in a microplate reader (Biotek Synergy 2). The percentage of viable cells for each drug concentration was determined by comparing the absorbance of the treated cells to the untreated control cells after subtraction of the corresponding blank.

### 2.4. MCT1, MCT4 and CD147 Expression Assessment

For the preparation of cell suspensions, cells were cultured in a complete growth medium in six-well plates. After reaching confluence, the medium was recovered, and cells were washed with PBS. The cells were incubated in a lysis buffer (50 mM Tris/HCl, pH 7.5, 30 mM NaCl, 0.5% Triton X-100, 1 mM EDTA.Na, 1 × protease inhibitor cocktail) for 20 min on ice and then centrifuged at 13,000 rpm for 5 min at 4 °C. After that, proteins were quantified with the Pierce BCA Protein Assay Kit (Thermo Scientific, Waltham, MA, USA), using bovine serum albumin as standard.

MCT1, MCT4 and CD147 levels were analyzed by Western blot, according to conventional procedures. Briefly, 20 μg protein were separated by sodium dodecyl sulphate polyacrylamide gel electrophoresis (SDS–PAGE) on a 7.5–10% polyacrylamide separating gel and transferred to a nitrocellulose membrane (Trans-Blot^®^ Turbo Blotting System, Bio-Rad, Hercules, CA, USA). After transfer, membranes were blocked with 5% non-fat dried milk in TBST (10 mM Tris/HCl, pH 7.5, 150 mM NaCl, and 0.2% Tween 20) at room temperature for 1 h. Membranes were incubated with the primary anti-MCT1 (diluted 1:100, Santa Cruz Biotechnology, Dallas, TX, USA), anti-MCT4 (diluted 1:1500, Santa Cruz Biotechnology) and anti-CD147 (diluted 1:100, Santa Cruz Biotechnology) antibodies overnight at 4 °C. α-Tubulin (diluted 1:200, Abcam, Cambridge, UK) was used as a loading control. Membranes were then incubated for 1h at room temperature with peroxidase-conjugated secondary antibodies (diluted 1:1500 in TBST with 1% non-fat dried milk) and washed 3 times for 10 min. Bands were visualized by treating the immunoblots with enhanced chemiluminescence (ECL) reagents and analyzed with The Discovery Series^TM^ Quantity One^®^ 1-D analysis software, version 4.6.5 (Bio-Rad). The protein content was evaluated by measuring the density of each band and normalizing it against the respective α-tubulin content.

### 2.5. Metabolic Assays

To study the effect of the GIs on the metabolism of lung cancer cells, extracellular glucose and lactate and intracellular ATP were quantified. Cells were incubated overnight in 96-well plates and then treated with the respective IC_50_ of each GI for 24 h in the case of 3BP and DCA, or 48 h for 2DG. For lactate and glucose determination, aliquots of 10 μL of the culture medium were collected and the metabolites quantified using commercial kits (Spinreact), according to the supplier’s instructions, and normalized against the respective total biomass, assessed by the SRB assay. For each metabolite, three different independent experiments were conducted in triplicate.

For ATP assays, the cells of the same wells were used, and intracellular ATP was measured by a bioluminescence assay using a commercial kit (Molecular Probes—Invitrogen), according to the manufacturer’s instructions. The ATP content was expressed as total ATP normalized against protein content, determined through the Pierce BCA Protein Assay Kit (Thermo Scientific, Waltham, MA, USA).

### 2.6. Proliferation Assay

The bromodeoxyuridine (BrdU) cell proliferation assay is an immunoassay for the quantification of BrdU, which is incorporated into newly synthesized DNA during cell proliferation. A549 and NCI-H460 cells were incubated overnight in 96-well plates and then treated with the respective IC_50_ of each GI for 24 h in the case of 3BP and DCA, or 48 h for 2DG. After treatment, cells were incubated with BrdU labeling solution, according to the manufacturer’s protocols (BrdU Cell Proliferation ELISA kit, Roche Applied Sciences). After labeling, the culture medium was removed and cells were fixed in FixDenat solution, which induces DNA denaturation. Then, the cells were incubated with the anti-BrdU-POD antibody for 90 min at room temperature. The antibody was removed, and the substrate solution was added to the washed cells. The reaction product was quantified by measuring the absorbance in a microplate reader (Biotek Synergy 2) at 370 nm, with a reference wavelength of 492 nm. Color development, and thereby the absorbance values, directly correlated with the number of proliferating cells in each specific condition.

### 2.7. Wound-Healing Assay

The effect of the GIs on cell migration was assessed by the in vitro wound-healing assay, which mimics the in vivo cell migration that occurs during wound-healing or cancer metastasis. 1.0 × 10^6^ A549 or NCI-H460 cells, corresponding to 2 mL cell suspension, were seeded in individual wells and incubated until total confluence was reached. At this point, the medium was removed, and two wounds were created in the confluent cells by manual scratching with a 200 μL pipette tip. Cells were then treated with FBS-free media containing each GI at the respective ½ IC_50_ for 24 h. Untreated cells were used as controls. At 0 and 24 h, the wound areas were photographed at 100× magnification using an inverted microscope (Eclipse TE 2000-U, Nikon, Tokyo, Japan). The relative migration distances of treated cells compared to the time zero of the control were analyzed using the ImageJ Software (version 1.52q).

### 2.8. Cell Death Assay

The Annexin V-Fluorescein isothiocyanate (FITC) and propidium iodide (PI) assay and caspase-3 activity determination were performed to assess the presence of GI-induced apoptosis and/or necrosis.

The Annexin V-FITC and PI assay were performed using the annexin V-FITC Detection Kit (Biotool) according to the manufacturer’s protocols. Cells were incubated overnight in six-well plates and then treated with the respective IC_50_ of each GI for 24 h in the case of 3BP and DCA or 48 h for 2DG. After incubation and trypsinization, the medium and the cells were recovered, the cells were washed with cold PBS and collected by centrifugation. The cells were re-suspended in Binding Buffer and incubated with Annexin V-FITC and PI for 15 min at room temperature. The percentage of viable, apoptotic and necrotic cells was assessed by flow cytometry (BD Accuri C6 Plus flow cytometer) with a total of 20,000 events, and the results were analyzed using the BD Accuri C6 Plus software (version 1.0.27.1).

Caspase-3 activity was analyzed, as described by Barbosa et al. [46,47], to confirm the type of cell death, since caspase-3 is one of the key effector enzymes involved in the apoptotic pathway. Cells were incubated overnight in six-well plates and, after incubation with the respective IC_50_ of each GI for 24 h (3BP and DCA) or 48 h (2DG), the medium was removed and the cells were washed with PBS. Then, the cells were incubated with 150 μL of Glo Lysis Buffer (Promega) for 5 min at room temperature and cell lysates were collected. In 96 well-plates, 50 μL of each lysate was mixed with 200 μL assay buffer (100 nM HEPES pH 7.5, 20% glycerol, 5 mM DTT, 0.5 mM EDTA) and 5 μL of the caspase-3 Ac-DEVD-pNA peptide substrate (Sigma-Aldrich) at a final concentration of 80 mM, followed by incubation at 37 °C for 24 h. The activity of caspase-3 was determined at 405 nm, by quantifying the reaction product in a microplate reader (Biotek Synergy 2), being further normalized against protein content.

### 2.9. Effect of DCA Pretreatment Cell on Paclitaxel Toxicity

Cells in the exponential growth phase were plated in 96-well plates and incubated overnight. After cell adhesion, the medium was removed and replaced by fresh medium with DCA at concentrations corresponding to ½ IC_50_ or IC_50_ values. After 24 h, the DCA-containing medium was removed, and after washing twice with PBS, the cells were exposed to a series of PTX concentrations for 48 h. As a control, a plate with a DCA-free medium was used and further processed in the same way. Cell viability was determined by the SRB assay.

### 2.10. DCA-Loaded PLGA NPs Formulation

PLGA NPs were produced by the water–oil–water (*w*/*o*/*w*) double emulsion technique, as described before [48,49]. Briefly, 18 mg of PLGA were dissolved in 900 μL of dichloromethane and 2 mg of PLGA-b-poly(ethylene glycol) (PLGA-PEG) dissolved in 100 μL of ethyl acetate, and both solutions were mixed. PLGA is one of the best characterized biodegradable and biocompatible copolymers that breaks down into non-toxic products (H_2_O and CO_2_) that are eliminated from the body [50,51]. Surface modification with PEG (PEGylation) increases the formulation hydrophilicity, as well as physiological stability against undesired aggregation and premature elimination [51]. Then, 2 mg of DCA were added and the solution was sonicated for 30 s, using a Vibra-Cell^TM^ ultrasonic processor at 70% amplitude, forming the first emulsion (*w*/*o*). After that, 4 mL of Pluronic^®^ F127 in ultrapure water were added and the solution was sonicated under similar conditions. Pluronic^®^ F127 is a surfactant polyol used to further stabilize the colloidal dispersion of PLGA NPs, and adjust the formulation parameters regarding desired size range. The second emulsion formed (*w*/*o*/*w*) was developed after the addition of 7.5 mL of the Pluronic^®^ F127 solution and the formulation was left under magnetic stirring at 300 rpm for 3 h to allow the evaporation of the organic solvent.

### 2.11. Characterization of DCA-Loaded PLGA NPs

The mean particle size and surface charge of the NPs were measured through the dynamic light scattering (DLS) method and electrophoretic light scattering (ELS), respectively, using the Malvern Zetasizer Nano ZS instrument (Malvern Instruments, Malvern UK). NPs were diluted 1:100 in a 10 mM sodium chloride (NaCl) solution at pH 7.4.

The Association Efficacy (AE) was calculated by an indirect method, where the amount of DCA encapsulated into PLGA NPs was calculated as the difference between the total amount of DCA used in the NP formulation and the free DCA in the supernatant. The AE was determined using the following equation: AE (%) = [(Total amount of DCA − Free DCA in supernatant)/Total amount of DCA)] × 100. The DL (drug loading) was calculated taking into account the total dry weight of PLGA NPs using the following equation: DL (%) = [(Total amount of DCA − Free DCA in the supernatant)/Total dry weight of NPs] × 100.

Free DCA in the supernatant was quantified by high-performance liquid chromatography (HPLC) in a Shimadzu UFLC Prominence System equipped with two LC-20AD pumps, a SIL-20AC autosampler, a CTO-20AC oven, a DGU-20A degasser, a CBM-20A system controller and a LC solution version 1.25 SP2. The UV detector was a Shimadzu SPD-20A, and the column used was a LiCrospher^®^ 100 RP-18 (5 mm) (250 mm × 4.6 mm) (Merck). Chromatographic analysis was performed in an isocratic mode where the mobile phase consisted of 5% acetonitrile and 95% of 2% phosphoric acid in ultrapure water. The eluent flow rate was 1.0 mL/min. The column was maintained at room temperature, and the injection volume was 20 μL. Detection was performed by UV at 214 nm. All samples were run in triplicate, and the total area of the peak was used to quantify DCA.

### 2.12. Effect of DCA-Loaded PLGA NPs on Cell Viability

The cells were seeded in 96-well plates as described and, after adhesion, incubated for 24 h with medium containing DCA-loaded PLGA NPs or free DCA at different concentrations (10; 50; 75; 100 and 125 μg/mL). The influence of empty PLGA NPs on cell viability was also tested and adjusted according to the concentrations of DCA-loaded PLGA NPs. At least three independent assays were performed in triplicate, and cell viability was determined by the SRB assay, assuming 100% viability for untreated cells in each case.

### 2.13. Statistical Analysis

The results presented correspond to the average of triplicates of at least three independent experiments. Results were expressed as means ± SD. For the statistical analysis, GraphPad Prism 8.3.1 software was used. All the assays were analyzed using one-way ANOVA, considering significant values to be * *p* < 0.05, ** *p* < 0.01, *** *p* < 0.001 and **** *p* < 0.0001.

## 3. Results

MDR is one of the major causes of treatment failure in cancer. This phenotype can be associated with several causes, including the energetic metabolism of cancer cells, which mainly relies on glycolysis, either in aerobic or anaerobic conditions. In this study, we aimed to exploit the effect of GIs on cancer cell properties, namely by investigating how they can overcome such a phenomenon of resistance to conventional drugs, opening doors for new therapeutic strategies. Thus, our overall objectives were: (i) to analyze the effect of the GIs 3BP, DCA and 2DG on lung tumor cell line properties; (ii) to verify their ability to reverse the MDR phenotype, when used in combination with PTX, a conventional drug used in lung cancer therapy; and (iii) to increase the efficiency of DCA delivery to tumor cells after its encapsulation into polymeric NPs, which may overcome limitations regarding the maximum dose of GIs that can be used.

### 3.1. 3BP, DCA and 2DG Decrease Lung Cell Viability in a Dose-Dependent Way

As a first approach to evaluate the toxic effect of GIs on the different lung cancer cell lines, namely A549 and NCI-H460, and on a noncancerous cell line derived from human pulmonary alveolar epithelial cells, HPAEpic, we determined cell viability after incubation with a range of concentrations of each GI (DCA, 3BP or 2DG), and determined the respective IC_50_, using the SRB assay (Table 1).

We observed that 3BP, DCA and 2DG decreased cell viability in all cell lines in a dose-dependent way. The NCI-H460 cancer cell line was shown to be the most sensitive to all three GIs. However, the other lung cancer cell line, A549, was shown to be the most resistant to 3BP and 2DG, whereas the normal cell line HPAEpic showed intermediate IC_50_ values for these GIs but a higher IC_50_ value for DCA (Table 1). A549 resistance to 3BP has already been mentioned in previous research, where the basal expression level of the 3BP target, HKII (which is also a target for 2DG), was reported to be very low [52].

### 3.2. MCT1, MCT4 and CD147 Basal Expression Is Not Correlated with the GIs Effect

Previous studies have demonstrated the contribution of MCTs to the absorption and toxicity of 3BP [53]. In order to understand MCT1 and MCT4’s influence on the effect of this GI, but also of DCA and 2DG, as all of them can interfere with lactate (a substrate of both transporters) levels, the expression of MCT1 and MCT4, as well as of their chaperone CD147, was quantified, having the expression levels in the non-tumor cell line HPAEpic as a reference (Figure 1).

Both the MCT1 and MCT4 transporters and CD147 were expressed in all cell lines, including the control one. In NCI-H460, the most sensitive cell line to the GIs assayed, a lower expression of the MCT4 protein has been found. In contrast, A549 cells, corresponding to the most resistant cell line to the GIs 3BP and 2DG, presented a higher MCT4 expression. No significant differences were observed in MCT1 expression, and the observable differences in MCT4 did not correlate with differences in the effect observed for the GIs. It could be expected that NCI-H460 cells, less resistant to all GIs, namely to 3BP, would present higher expression of its transporter MCT1 and/or of the respective chaperone, or even of MCT4, which has also been reported to be involved in 3BP uptake, but this was not observed. Therefore, these results indicate that other factors should contribute to the different sensitivity to the drugs. In fact, as aforementioned, the most resistant cell line, A549, was reported to have low expression of the main 3BP target (and also of 2DG), HKII [52]. Furthermore, the lower MCT4 expression in the NCI-H460 cell line could lead to a lower lactate efflux, inducing an increase in intracellular acidification and in cell death. Although other reports described the influence of both MCT1 and MCT4 in GIs effect [54], this seems to not be the case for these cell lines.

### 3.3. 3BP, DCA and 2DG Induce Cell Death, Both by Apoptosis and Necrosis

To assess the cell death mechanism induced by the GIs 3BP, DCA and 2DG, Annexin V/PI and caspase-3 assays were performed. Figure 2 shows the results concerning the Annexin V/PI assay in cells treated with the respective IC_50_ values of each GI. Flow cytometry analysis showed that the mechanism of cell death depended on the cell line and on the compound used.

Concerning the A549 cell line, untreated cells (control) showed a basal level of around 0.5% and 10% of apoptotic and necrotic cell death, respectively. Both DCA and 2DG, but not 3BP, induced an increase in apoptotic levels, whereas necrosis was stimulated by DCA and 3BP but not by 2DG. DCA induced the greatest effect, resulting in 40% of cell death, mainly by necrosis (approximately 36 ± 4.31% of necrosis and 3% of apoptosis). The treatment with 2DG induced an increase in apoptotic cells only (around 3%) and 3BP in necrotic cells only (around 20%).

An increase in caspase-3 activity was not observed in A549 cells treated with 3BP, DCA or 2DG (Figure 3), which might indicate that probably apoptosis was not the main mechanism responsible for cell death, in agreement with the results produced by the Annexin V/PI assay for 3BP and DCA. Concerning 2DG, although an increase in apoptotic cells was detected by annexin staining, this effect was very small and not reflected in caspase-3 activity.

Concerning the other cell line, NCI-H460, untreated cells (control) showed a basal level of around 10% and 4% of apoptotic and necrotic cell death, respectively. In this case, and differently from A549 cells, DCA induced apoptosis only, whereas 3BP induced both mechanisms of cell death. No effect, both in apoptosis and necrosis, was observed in 2DG-treated cells. In this cell line, it was 3BP that produced the greatest effect, resulting in 70% of total cell death (45% of apoptosis and 25% of necrosis), whereas DCA significantly induced an average of 38% of total cell death (about 35% of apoptosis and 3% of necrosis). In agreement with these results, treatment with 3BP revealed an increase in caspase-3 activity. However, the increase in apoptotic cells, determined by the annexin assay for DCA-treated cells, was not confirmed, suggesting that the annexin assay is more sensitive than the caspase-3 assay. For 2DG, no increase in apoptotic rate was observed, neither through the annexin assay nor through the caspase-3 activity assay. As such, these results may suggest that 2DG induces cell death by another mechanism—likely autophagy. In fact, some authors have reported that, in vitro, 2DG induces autophagy in different tumor cell types [55,56].

In summary, for both cell lines, an increase was observed for both apoptosis and necrosis, depending on the cell line, mainly upon 3BP and DCA treatment. Our results are in agreement with other reports that also showed that 3BP induces apoptosis and necrosis and that DCA induces mainly apoptosis [29,57]. In fact, GI-induced ATP depletion can be a major factor in cell death. Concerning 3BP, the inhibition of HKII increased the mitochondrial permeability and thus the release of cytochrome C, activating caspases that induce apoptosis [57]. Regarding DCA, and since it is a molecule that can reverse the Warburg effect, the stimulation of oxidative metabolism may have caused an increase in ROS production, with mitochondrial overload and, consequently, the induction of cell death. In fact, such overload can result in impaired efficiency of antioxidant defenses, which will be unable to cope with the excessive amount of ROS [29].

### 3.4. DCA Is the Glycolytic Inhibitor with Greater Effect on the Metabolism of Lung Cancer Cells

In order to understand if the effect of GIs on cell viability was due to metabolic disturbance in cancer cells, glucose consumption and lactate production, as well as ATP levels, were assessed in A549 and NCI-H460 cell lines (Figure 4).

GIs exposure is expected to lead to a decrease in lactate production and glucose consumption, causing cellular ATP depletion and, consequently, cell death [58,59,60]. A549 and NCI-H460 were treated for 24 h (in the case of 3BP and DCA) or 48 h (in the case of 2DG) with IC_50_ values of 3BP, DCA and 2DG. After treatment, extracellular glucose and lactate and intracellular ATP were quantified and normalized against total biomass or protein, respectively (Figure 4). As expected, in treated cells, we observed, in general, a decrease in glucose consumption and in lactate and ATP production, with this effect being more evident when DCA was used.

Firstly, glucose consumption was shown to decrease after GIs 3BP and DCA treatment in both cell lines, except for A549 cells treated with 3BP, in which the decrease was not significantly different compared to the control. Nevertheless, the most pronounced effect was observed in NCI-H460 cells treated with DCA. Concerning 2DG treatment, extracellular glucose was not determined because, since this compound is a glucose derivative, it reacts with the colorimetric reagent, making it impossible to quantify it with the method used. In turn, as far as lactate production is concerned, the results confirm the effect of GIs on glycolysis, as lactate levels are reduced in both cell lines, except for NCI-H460 cells treated with 3BP. Regarding ATP production, in the A549 cell line, only DCA was able to significantly reduce it, whereas, in the NCI-H460 cell line, there was a decrease after treatment with 3BP and DCA.

In the most resistant cancer cell line, A549, the amount of glucose consumed after inhibition with DCA was reduced almost by half (from 60 mg/dL to 35 mg/dL, approximately) and, in the most sensitive cell line, NCI-H460, this reduction was even more evident (from around 55 mg/dL to 10 mg/dL). DCA treatment also led to the depletion of cellular ATP in both lung cancer cell lines, with a decrease to less than half in A549 and a decrease of approximately 40% in NCI-H460. Accordingly, DCA treatment lowered lactate levels in both cell lines: in A549 cells, lactate produced was reduced from 30 mg/dL (control) to approximately 13 mg/dL (treated cells) and, in NCI-H460 cells, from 35 mg/dL (control) to 10 mg/dL (treated cells). These results indicate that glucose oxidation switched from fermentative glycolysis toward oxidative mitochondrial metabolism. Since DCA can reverse the Warburg effect through PDH activation, DCA-induced stimulation of oxidative metabolism interrupts the metabolic advantage of tumor cells. Due to the frequent occurrence of mutations in their mitochondrial DNA, tumor cells often present dysfunction of the respiratory chain. As a result, they become unable to sustain energy demand [29]. Furthermore, by decreasing lactate production, DCA neutralizes the acidosis state of the tumor microenvironment, which can contribute to the inhibition of tumor growth.

Different results were obtained when the cell lines were treated with the GI 3BP. As previously noted, the A549 cell line was less sensitive to this compound, with a non-significant reduction of the glucose consumed and ATP cell content. However, a significant decrease in lactate production was observed, similarly to the DCA treatment. Consistently with the results that indicate that the A549 cell line has higher rates of oxidative metabolism, the 3BP treatment did not affect the energetic yield of this cell line. In fact, there can be cases where cancer cells also rely on oxidative metabolism. Moreno-Sanchez described the contribution of OXPHOS in a model of lung cancer, where the majority of ATP was produced during OXPHOS [61,62]. This means that OXPHOS might serve as an additional rescue energy alternative in these cells when glycolysis is inhibited. 

In NCI-H460, the glucose consumed (50 mg/dL, approximately in control cells) was decreased by half after treatment with 3BP, while ATP production was significantly reduced to less than half. However, this alteration was not accompanied by a decrease in lactate production. Due to the metabolic plasticity exhibited by tumor cells, it is not unexpected that these cells could develop resistance to inhibition of a specific pathway through the upregulation of alternative pathways [61]. It is known that energy production in tumor cells, in addition to glucose oxidation, is mediated by glutamine metabolism. Glutamine is essential for tumor cells as the amine group is critical for the biosynthesis of other molecules, and important for tumor proliferation [63,64]. In this sense, glutamine-derived glutamate will be a precursor of pyruvate. However, due to modified metabolism, cancer cells frequently convert pyruvate into lactate rather than into acetyl-CoA, contributing to an increase in lactate levels [63,65].

When 2DG was used, the ATP content was not reduced in both cell lines. However, this GI was shown to be able to decrease lactate production, both in A549 and NCI-H460 cells, which is in agreement with its inhibitory effect on the glycolytic pathway.

These results show that lung cancer cell lines treated with GIs, namely with DCA, suffer a disruption in their metabolism, with a significant decrease of energy, particularly the NCI-H460 cell line, which is also the most sensitive to the drugs. Furthermore, DCA was the only GI capable of disturbing ATP production in the most resistant cancer cell line, A549.

### 3.5. DCA Decreases Proliferation of Lung Cancer Cells

To verify the role of metabolic inhibition on cell proliferation, the BrdU assay was performed on cells treated with the IC_50_ value of 3BP, DCA (24 h) or 2DG (48 h) (Figure 5). The treated cells were then cultured in a medium containing BrdU, with this pyrimidine analog being incorporated instead of thymidine into the newly synthesized DNA in dividing cells. After DNA denaturation, the incorporated BrdU was detected by labeling it with the respective antibody.

As shown in Figure 5, we observed that 2DG, contrary to expectations, induced proliferation in A549 cells. It can be seen that, in the metabolism assay, no inhibition by 2DG was observed on ATP production. In this case, the proliferation was not inhibited, as the opposite occurred. This can be attributed to the fact that, in some types of tumors, the efficacy of 2DG is limited because glycolytic enzymes are overexpressed, and consequently, the concentration of 2DG used may not be sufficient to have an effect on the parameters analyzed [56]. Furthermore, its success as a GI is described as controversial, as this compound was found to activate multiple pro-survival pathways in tumor cells [61].

In both cell lines, the highest effect was observed for DCA, where 50% of cell proliferation was inhibited. In effect, the use of glucose supplies cells with intermediates used in other pathways, like lipid, nucleotide and amino acid biosynthesis [66,67]. As such, the decrease in metabolism will lead not only to a decrease in ATP, essential for cell proliferation, but also in glycolytic intermediates, such as glucose-6-phosphate, which can fuel the pentose phosphate pathway, thus decreasing the availability of biosynthetic intermediates [67]. In addition, the reduction in lactate production promoted by DCA also had consequences on cell proliferation. Lactate produced by glycolysis in tumor cells is taken up by neighboring cells and converted into pyruvate, which enters the mitochondria of aerobic cells to be used in OXPHOS, generating ATP. Such lactate transport allows not only tumor growth but also the inhibition of cell death mechanisms [29,61].

### 3.6. DCA Decreases Migration of Lung Cancer Cells

Migration is a process that offers valid targets for intervention in important physiological and pathological phenomena, such as wound healing and cancer metastases [68]. To study the effect of GIs on cell migration, the in vitro wound-healing assay, based on the healing process with the aim of mimicking the ability of cells to migrate in vivo, was performed, and the migration of tumor cells was registered at 0 and 24 h [69].

The NCI-H460 cell line exhibited a slightly higher migratory capacity (Figure 6). However, both cell lines exhibited a low migratory capacity, and consequently, the GIs did not have a major impact on their migration. 3BP was the only GI that affected migration, and only in NCI-H460 cells, in which a decrease of around 41% was observed with ½ IC_50_ of 3BP. Therefore, the results suggest that 3BP seems to influence the migratory capacity of cells, and such ability may contribute to its anticancer effect. In the A549 cell lines, again, an increase in cell migration was unexpectedly observed with 2DG treatment, consistently with its effect on cell proliferation.

### 3.7. DCA Increases the Sensitivity of Lung Cancer Cells to Paclitaxel

PTX is one of the most commonly used anticancer drugs in therapies against solid tumors, although the disease relapses frequently due to the development of resistance to the drug [70]. Such resistance has been attributed to a decrease in drug accumulation within the cell, mainly due to an overexpression of protein efflux pumps that have PTX as substrate, from which Pgp is one of the most important [70,71,72]. Its overexpression, as well as that of other efflux pumps, contributes to the MDR phenotype, namely in lung cancer [71,72,73]. However, MDR is also the biological result of cellular adaptation to conditions that include microenvironmental changes due to its reprogrammed metabolism, such as hypoxia, acidosis or nutrient deficiency. Therefore, metabolic inhibition can result in modifications of these microenvironmental features, also involved in MDR [41]. In this way, determining adjuvant therapies that could interfere with metabolism and inhibit the MDR phenotype may increase lung cancer cell line sensitivity to chemotherapy.

Cells expressing MDR proteins, such as Pgp, are known to require ATP as the energy source to pump out drug substrates [74]. Thus, inhibition of the main energy production pathways in tumor cells may cause a decrease in drug efflux due to cellular ATP depletion, which may contribute to decreased drug resistance [41]. Since DCA was the most promising GI inhibiting metabolism in the assayed cancer cells, we analyzed the effect of this GI on the MDR phenotype. For that, cells were first exposed to DCA and then treated with PTX. Furthermore, since DCA is an inhibitor of PDK, an enzyme with low expression in normal tissues, the use of DCA may spare healthy cells, minimizing adverse effects [61]. In this sense, and in order to clarify the combinatorial effect on normal cells, the HPAEpic cell line was used in this assay (Table 2).

The three untreated cell lines presented similar sensitivity to PTX and, in all of them, the IC_50_ value decreased when the cells were pre-incubated with DCA, showing that this GI can sensitize cells to PTX. However, this effect is less evident in the nontumor cell line HPAEpic. In fact, although this cell line is more sensitive than the A549 cell line to two of the GIs studied, the effect of potentiation appears to be more specific in tumor cell lines. In contrast, for both cancer cell lines, such effect was very evident, even with a lower concentration of DCA (½ IC_50_), but much more pronounced in the NCI-H460 cell line. In A549, the most resistant cell line to PTX and to GIs, the IC_50_ value decreased 2.7-fold, whereas, in the NCI-H460 cell line, the IC_50_ value decreased 10.1-fold. ATP depletion and exported lactate should affect proteins putatively involved in chemoresistance, which can be present in cancer cells. In fact, the cancer cell line treatment with DCA had almost the same effect on metabolism in both cell lines (as assessed through cellular ATP levels and lactate production), for which a similar effect could be expected for the decrease in PTX IC_50_ in both cell lines after DCA pretreatment. However, the decrease in PTX IC_50_ did not parallel the effects on metabolic parameters, being more pronounced in NCI-H460 cells. These were shown to be intrinsically more sensitive to GIs, which suggests that other metabolic parameters and/or membrane transporters and proteins involved in drug resistance may contribute to cell line sensitivity to PTX.

### 3.8. DCA-Loaded NPs Decrease Cell Viability

Our results demonstrate that the biological activity of DCA is mainly due to its ability to decrease tumor cell metabolism. However, there are disadvantages to a metabolism-based approach in cancer therapy, since the metabolic pathways required for cell survival are also present in normal cells. Thus, metabolism-based treatment can face a major hurdle of non-specific toxicity [61]. Therefore, to increase cellular internalization of DCA by tumor cells, thereby increasing its specific anticancer activity with lower side effects, its nanoencapsulation was performed. In fact, encapsulation of DCA into nanocarriers holds the potential to increase its delivery into the cell, where the target components are present (e.g., PDH, PDK), thus requiring a smaller amount of the compound to elicit therapeutic effects. Furthermore, PLGA is a polymer that has been extensively explored for the development of controlled drug delivery systems of small drug molecules [75]. In this study, we formulated DCA-loaded PLGA NPs through the double emulsion technique since it offers, in most cases, high encapsulation/association efficiency and a controlled release [76]. The physicochemical properties of empty PLGA NPs and loaded PLGA NPs are described in Table 3.

The average size is within the 125–130 nm range. The encapsulation of DCA into PLGA NPs did not change the particle size. However, there was an increase in the polydispersity index (PdI, from 0.099 to 0.183) and a decrease in the zeta potential (from −4.31 to −8.99). The increase in PdI and the decrease in the zeta potential may be indicative of some aggregation and fusion of nanoparticles [77]. However, it is well known that, for a homogenous NP suspension, the PdI should be below 0.2, meaning that all our formulations are stable and homogenous [78]. The negative charge of PLGA NPs is associated with the negative charge of PLGA [79]. Since our formulation was considered stable, we assessed cell viability upon exposure to PLGA NPs, through the SRB assay, as previously described.

The formulations, represented as DCA and DCA-loaded PLGA NPs, showed a concentration-dependent effect on the viability of lung cells. After 24 h, no significant differences were found in all samples at concentrations between 10 and 100 mg/mL. Viability results showed that PLGA NPs did not lead to significant cytotoxicity in lung cells, which is in agreement with the literature [80]. However, it is well known that the highest biological effect of DCA is achieved with high concentrations, for which we also performed the viability assay with a concentration as high as 125 mg/mL (Figure 7).

Even so, this concentration is much lower than those used in the previous assays. The results showed that DCA-loaded NPs allowed a decrease in cell survival (*p* < 0.01 for A549 and NCI-H460 cell lines and *p* < 0.001 for the HPAEpic cell line), which might be interpreted as a result of increased intracellular deposition of the drug. The decrease in viability might be associated with the fact that DCA has been internalized by lung cancer cells, binding to the intracellular components, namely PDK. By blocking PDK, DCA shifts pyruvate metabolism from glycolysis to OXPHOS, allowing a decrease in cell viability. Indeed, our study demonstrated that encapsulation was successfully achieved, enabling an observable biological effect on lung cancer cells. Although not explored here, it may be expectable that the surface modification of NPs with an active targeting fraction would increase target specificity [81]. Future work is being planned regarding functionalization for EGFR targeting ability.

## 4. Discussion

Although conventional chemotherapy is particularly toxic to tumor cells, it is often non-specific, being responsible for the significant side effects associated with cancer treatment. However, there are differences between tumor cells and healthy cells that can be explored to increase treatment specificity against cancer. One of these differences consists of the Warburg effect, which is currently considered a new cancer hallmark, whereby the upregulation of the glycolytic rate in tumor cells is a key player in acid-resistant phenotypes by adaptation to hypoxia and acidosis, and in tumor aggressiveness [3,4,5]. Exploring specific characteristics of tumor cells, such as this change in metabolism, could be a promising strategy for the use of more effective and specific drugs that primarily target tumor cells. That is the case for several GIs developed over the last years, such as 3BP, DCA and 2DG. 

Since anticancer drugs often decrease the proliferative capacity of the cells, affecting cell death and migration capacity, the effect of GIs was assessed in this context, as well as in the metabolic status of the cell, according to their primary function. Thus, a series of experiments were performed in this work, aiming to understand the effect of 3BP, DCA and 2DG in cancer, using lung cells as a model. Furthermore, due to previous findings in human cancer tissues, we aimed to dissect the association of MCTs with the GIs effect. Although MCTs are reported to have a contribution to the toxicity of some GIs, namely 3BP [53], in the present work, the expression of MCT1 and MCT4, as well as of their chaperone CD147, was not correlated with the GIs effect. Concerning the GIs’ effect on cancer cell characteristics, in the A549 and NCI-H460 lung cancer cell lines, as well as in the non-tumor cell line HPAEpic, all the GIs assayed led to a decreased percentage of viable cells in a dose-dependent way, with the lung cancer cell line NCI-H460 being the most sensitive to all the compounds. As previously mentioned, 3BP and DCA have been used to target glycolysis, and 2DG to compete with glucose in the first step of its intracellular metabolism. Accordingly, in order to understand if the effect of GIs on cell viability was due to metabolic disturbance, glucose consumption and lactate and ATP production were measured in the lung cancer cell lines. Our results showed that GIs, in particular DCA, decreased lactate and ATP production and glucose consumption in the cell lines, confirming its inhibitory effect on glycolysis. Glucose consumption provides cells with the necessary intermediates for the lipid, nucleotide, and amino acid biosynthetic pathways. Furthermore, the lactate produced constitutes a substrate for oxidative tumor cells [15,82]. Nevertheless, in spite of the effect observed in cell metabolism, only a small effect was observed on the inhibition of the migratory capacity, except for 3BP in the NCI-H460 cell line. Migration is one of the major steps in the metastatic cancer cascade, through which cancer cells are able to become motile to escape the primary tumor and move to a different location. Our results showed that the cell lines assayed exhibit already intrinsically a low migratory capacity in basal conditions and, consequently, the GIs did not have a major impact on their migration. The anticancer effect of a compound is a balance between enhanced cell death and decreased cell migration and cell proliferation. In relation to this, we also studied the contribution of GIs to the inhibition of cell proliferation. DCA decreased cell proliferation in the cell lines under study, while, for the other GIs, the cells were more resistant to such inhibition. Self-sufficiency in growth factors and insensitivity to anti-growth factors are known to promote tumor cell proliferation, and there are multiple mechanisms by which constitutive activation of growth factor signals may be associated with metabolic reprogramming [83]. Inhibition of glycolytic activity had an inhibitory effect on cellular metabolism due to impairment of glucose consumption and lactate and ATP production, and this can also affect signaling pathways involved in cell proliferation. Consequently, the aggressiveness potential of these cells decreased through the inhibition of proliferation and migration and by the increase in cell death.

High glycolytic rates are widely reported to promote chemoresistance of tumor cells to conventional therapy [3]. In fact, increased acidification of the extracellular space leads to lower drug stability and, consequently, lower drug efficacy. In parallel, increased production of glycolytic intermediates promotes cell proliferation since these are biosynthetic precursors, whereas ATP production sustains both the activity of proteins involved in drug efflux and cell division. Together, these effects underlie multidrug resistance. Our results showed that the pretreatment with DCA made the cells more sensitive to the action of PTX, probably due to its effect on tumor cell metabolism, since it decreased the production of glycolytic intermediates, lactate and ATP. It should be emphasized that HPAEpic, as a normal cell line, is expected to have a lower PDK expression when compared with NCI-H460 and A549 cell lines, as well as a lower dependence on glycolysis. Therefore, the modulation/inhibition of the glycolytic metabolism via DCA pretreatment has a more pronounced impact on the tumor cell line sensitivity to the conventional anticancer agent PTX. This effect of DCA increasing the sensitivity to PTX in cancer cells was also reported by Zhou et al. [84]. The authors observed that lung cancer cell treatment with DCA restores the sensitivity to PTX in a PTX-resistant cell line derived from A549, defective in mitochondrial respiration. According to the authors, the effect of DCA inhibiting Pgp activity is more effective in cells with damaged mitochondria (A549/Taxol versus A549) and, thus, were unable to restore ATP production via OXPHOS. In these cells, the tricarboxylic acid (TCA) cycle cannot be activated, which can lead to the accumulation of intermediates of the TCA cycle. In fact, the authors observed greater levels of citrate accumulation in the A549/Taxol. Citric acid is an inhibitor of the glycolytic enzyme phosphofructokinase, having a crucial role in inhibiting the Warburg effect. 

In our work, we also observed this increase of sensitivity to PTX in A549 cells, as well as in the DCA more sensitive cell line NCI-H460. It is described that a metabolic switch to OXPHOS in cells expressing the wild-type p53 (like both cancer cell lines here used [85]), treated with DCA, induced a lower expression of the gene *ABCB1*, coding for Pgp, as well as of others efflux pumps [86]. This, together with the metabolic alterations, including ATP depletion, in these cells DCA-treated cells, can explain the increased sensitization to PTX observed.

Our results with DCA were quite promising, given that the decrease in cell viability upon DCA pretreatment was higher for the tumor cell lines than for the normal cell line. However, the fact that the effect on cell viability was not absent for HPAEpic underlines the need to enhance drug targeting to tumor cells. Since the inhibition of the metabolism of healthy cells and the significant drug accumulation outside the tumor cells could lead to serious adverse effects [51], we aimed to analyze the effect of DCA encapsulation on its delivery and toxicity to cancer cells. Our results demonstrated that the DCA-loaded NPs allowed for a decrease in cell survival compared to the free DCA. Although this was only observed at the highest concentration tested, this concentration was lower than all assayed in previous experiments. The results show that nanoencapsulation can be a promising strategy to increase the intracellular delivery of DCA and, thus, increase the inhibition of tumor cell metabolism.

Tumor cell biology is extremely complex, and an array of factors can be involved in the MDR phenotype, thus compromising chemotherapy outcomes. Many other components (e.g., transporters, metabolic substrates and intermediates), complementary to those assayed in this work, putatively represent valuable targets of anticancer therapies and are being explored as part of new therapeutic approaches. In this work, the effect of a standard drug already in use (PTX) has been intensified by exploring the reprogrammed metabolism as the ‘Achilles heel’ of cancer cells through the use of a GI (DCA). The effect was further potentiated by NPs encapsulating the DCA. Thus, the results herein presented demonstrate the potential of “all in one” therapeutic approaches, combining multiple strategies (glycolysis inhibition, microtubule dynamics modulation, nanoencapsulation) as the key to efficiently and selectively targeting tumor cells.

## Figures and Tables

**Figure 1 pharmaceutics-14-02021-f001:**
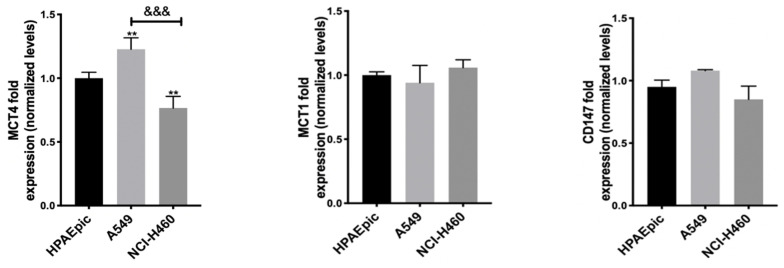
MCT1, MCT4 and CD147 expression analysis in HPAEpic, A549 and NCI-H460 cell lines, assessed by Western blot. The noncancerous cell line HPAEpic presenting a normal phenotype was used as a reference. Levels of protein expression are relative to the control cells and were normalized against tubulin. The results are presented as means ± SD of two independent experiments. ** *p* < 0.01 compared to HPAEpic cells (control). *^&&&^ p* < 0.001 compared to A549 cells.

**Figure 2 pharmaceutics-14-02021-f002:**
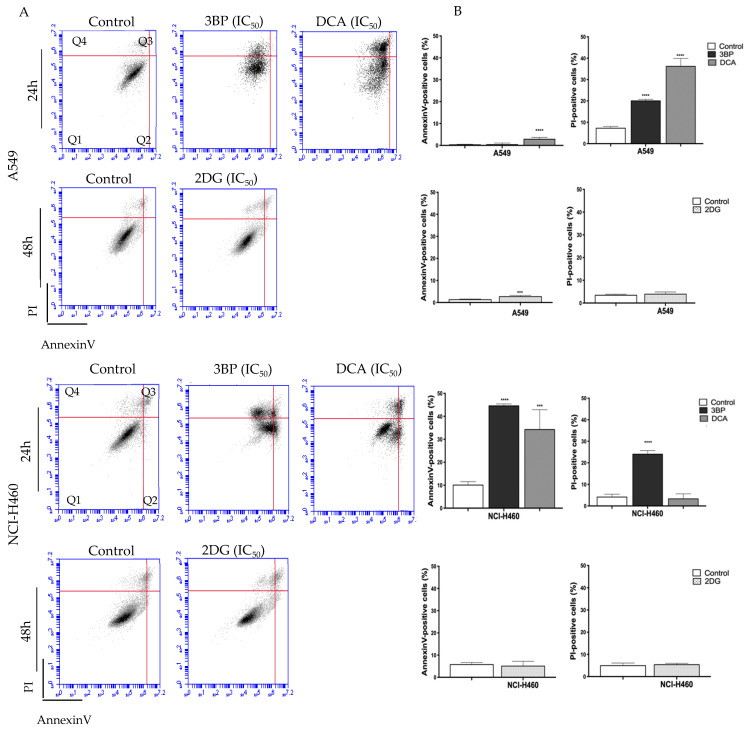
Effect of GIs on cell death after 24 h of treatment with 3BP and DCA or after 48 h with 2DG. Representative cytograms (**A**) and quantification of Annexin V- and PI-positive cells (**B**) are shown for A549 (top) and NCI-H460 (bottom) cell lines. The quadrants (Q) were defined as Q1 = live (Annexin V- and PI-negative), Q2 = early stage of apoptosis (Annexin V-positive/PI-negative), Q3 = late stage of apoptosis (Annexin V- and PI-positive) and Q4 = necrosis (Annexin V-negative/PI-positive). *** *p* < 0.001; **** *p* < 0.0001 compared to untreated cells (control).

**Figure 3 pharmaceutics-14-02021-f003:**
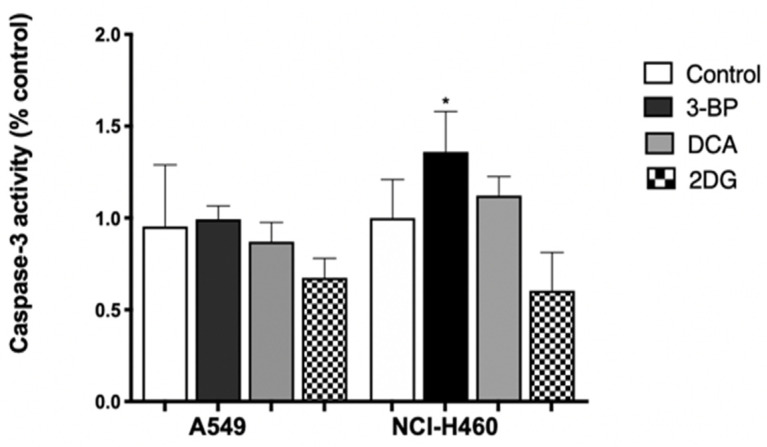
Effect of GIs on the caspase-3 activity of cells after 24 h of treatment with 3BP and DCA or after 48 h with 2DG. Quantifications were performed, normalizing the enzyme activity against the protein content of the extract and also against the value obtained in the absence of GIs. The results represent the mean ± SEM of at least three independent experiments. * *p* < 0.05 compared to untreated cells (control).

**Figure 4 pharmaceutics-14-02021-f004:**
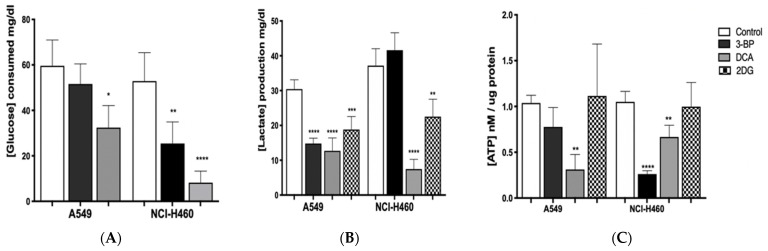
Metabolic profile of the lung cancer cell lines A549 and NCI-H460, estimated by (**A**) glucose consumption and (**B**) lactate and (**C**) ATP production, after treatment with GIs. Results are presented as means ± SEM, in triplicate, of at least three independent experiments. Significantly different between groups: * *p* < 0.05; ** *p* < 0.01; *** *p* < 0.001; **** *p* < 0.0001 compared to untreated cells (control).

**Figure 5 pharmaceutics-14-02021-f005:**
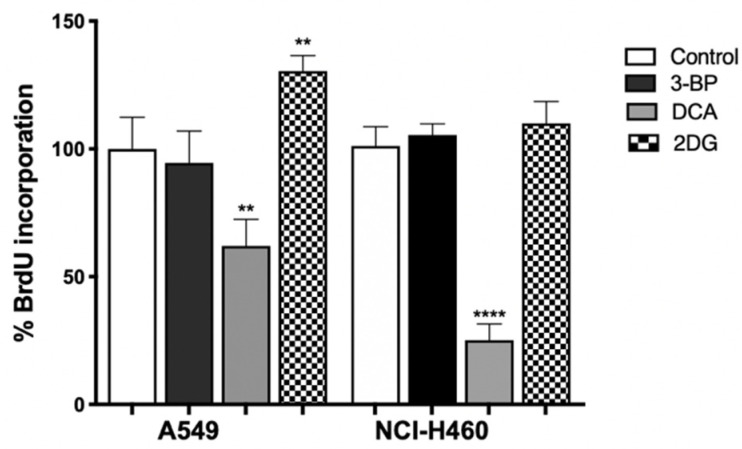
Effect of GIs on cell proliferation of lung cancer cells. The cell lines were treated with the respective IC_50_ of 3BP and DCA for 24 h and with 2DG for 48 h. Cell proliferation was assessed through the percentage of BrdU incorporated into the DNA of the treated cells. Results represent the mean ± SEM of a least three independent experiments, each one in triplicate. ** *p* < 0.01; **** *p* < 0.0001 compared to untreated cells (control).

**Figure 6 pharmaceutics-14-02021-f006:**
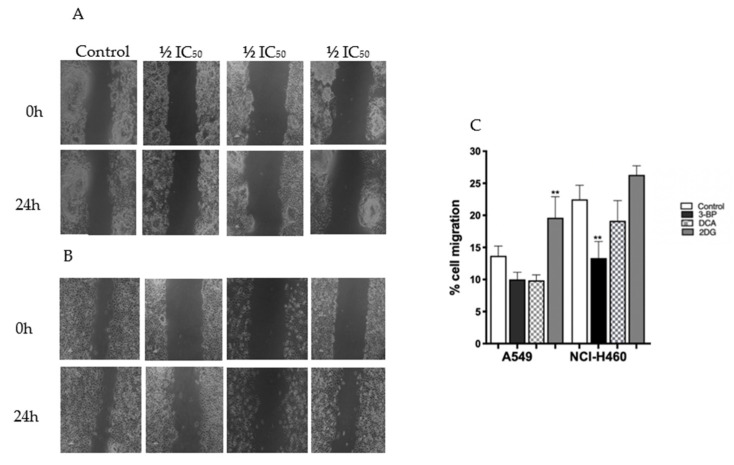
Effect of 3BP, DCA and 2DG at concentrations of 0 (control) and ½ IC_50_ on A549 and NCI-H460 cell migration (0 and 24 h of treatment) estimated by the wound-healing assay. (**A**,**B**) Photographic records of A549 and NCI-H460, respectively. (**C**) Quantitative results. Results represent the mean ± SEM of at least three independent experiments. Significantly different between groups: ** *p* < 0.01 compared to untreated cells (control).

**Figure 7 pharmaceutics-14-02021-f007:**
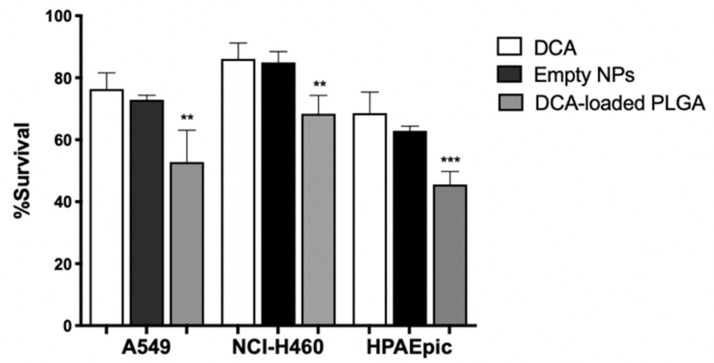
Effect of DCA, DCA-loaded PLGA NPs and empty NPs on cell viability of A549, NCI-H460 and HPAEpic cells. SRB assay of A549, NCI-H460 and HPAEpic cells treated with 125 μg/mL of DCA, DCA-loaded PLGA NPs, or empty NPs at 24 h. Results are expressed as means ± SD of triplicates from at least three independent experiments. ** *p* < 0.01; *** *p* < 0.001 compared to DCA (control).

**Table 1 pharmaceutics-14-02021-t001:** IC_50_ values of DCA, 3BP and 2DG for A549, NCI-H460 and HPAEpic cell lines.

Cell Line		GIs	
	3BP (μM)	DCA (mM)	2DG (mM)
A549	211.4 ± 11.5	24.6 ± 3.7	18.2 ± 7.2
NCI-H460	57.9 ± 15.6	12.7 ± 3.8	4.5 ± 0.5
HPAEpic	155.1 ± 7.4	42.8 ± 10.4	6.0 ± 2.2

**Table 2 pharmaceutics-14-02021-t002:** The effect of DCA pre-incubation in the IC_50_ values of PTX in A549, NCI-H460 and HPAEpic cell lines. The results are presented as means ± SD of at least three independent experiments. * *p* < 0.1; ** *p* < 0.01; *** *p* < 0.001; compared to cells without DCA (control).

		IC_50_	
Cell Line	A549 (mM)	NCI-H460 (mM)	HPAEpic (mM)
0 DCA + PTX	55.7 ± 1.8	50.6 ± 9.9	59.4 ± 2.4
½ IC_50_ DCA + PTX	25.6 ± 5.0 **	7.4 ± 4.3 ***	48.0 ± 8.7
IC_50_ DCA + PTX	20.5 ± 4.8 **	5.0 ± 1.3 ***	36.9 ± 11.0 *
PTX/DCA + PTX Index ^1^	2.7	10.1	1.6

^1^ Cells incubated for the same period of time in a DCA-free medium were used as control. The PTX sensitivity index was determined by comparing IC_50_ values of the control with the ones determined in cells exposed to DCA. Results are expressed as means ± SD of triplicates from at least three independent experiments.

**Table 3 pharmaceutics-14-02021-t003:** Physicochemical properties of unloaded NPs and DCA-loaded PLGA NPs.

Formulation	Z-Average (Size, nm)	Polydispersity (PdI)	Zeta Potential (mv)	AE (%)	DL (%)
Empty PLGA NPs	125.1 ± 0.2	0.099 ± 0.012	−4.31 ± 0.23	NA	NA
DCA-loaded PLGA NPs	130.1 ± 3.9	0.183 ± 0.019	−8.99 ± 0.61	33.0 ± 7	3.0 ± 1

## Data Availability

Data is contained within the article.

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
