# Peer review of "Glycolytic Inhibitors Potentiated the Activity of Paclitaxel and Their Nanoencapsulation Increased Their Delivery in a Lung Cancer Model"

_pharmaceutics, 2022, doi:10.3390/pharmaceutics14102021_

Round 1

Reviewer 1 Report

Concerning the mansucript entitled "Glycolytic inhibitors potentiated the activity of paclitaxel and their nanoencapsulation increased their delivery in a lung cancer model" by Cunha et al: This an excellently designed, analyzed and scientifically presented study showing that the nanoencapsulation of already potent inhibitors of glycolysis greatly enhances their anti-cancer activity via increased delivery to the cells. Interestingly, they found that DCA was the most effective of the substances that they studied.

While the scientific aspects and data presentation and conclusions were excellent and merit immediate publication, I would like the authors to give the manuscript to an English editor to improve as the presentation is quite stilted. These are little things but they will help the readability. Examples are in the abstract such as: "These compounds led to loss of cell viability, being their effect on cell metabolism, migration and proliferation 20 dependent on the drug and cell line assayed." which needs to be great altered ie: These compounds led to loss of cell viability, with different effects on cell metabolism, migration and proliferation depending on the drug and cell line assayed." The following sentences in the abstract also need to be corrected and this continues throughout the manuscript. After this English has been corrected for readability and understanding, I wholeheartedly recommend this manuscript for publication.

Author Response

Authors’ response to general comments: We thank the positive comments on our study. We are also very grateful for the attentive review made on our draft and all the valuable suggestions resulting from it. The remarks in the abstract, and in all manuscript, were carefully considered in our revision and corrected, hoping that the changes performed improved the readability and understanding of the manuscript, as well as its quality and consistence.

Reviewer 2 Report

This is an interesting study about glycolytic inhibitors potentiating the activity of paclitaxel with nanoencapsulation to increase their delivery. I recommend it for publication after the following minor points are fixed.

1. Drug loading content for different drugs should be calculated.

2. The authors should add explanations about the different roles of PLGA-b-poly(ethylene glycol)-maleimide and Pluronic® F127 during nanoparticle preparation.

3. Why were the zeta potential of nanoparticles negative?

4. In this study, targeting groups were not modified on the nanoparticles, why was PLGA-b-poly(ethylene glycol)-maleimide used but not PLGA-b-poly(ethylene glycol)?

5. In the introduction, 'Among polymeric NPs', other polymeric NPs (Langmuir 35 (5), 1273-1283; Colloids and Surfaces B: Biointerfaces, 189, 110830; Pharmaceutics 13.8 (2021): 1319) are encouraged to be included.

Author Response

General comments: This is an interesting study about glycolytic inhibitors potentiating the activity of paclitaxel with nanoencapsulation to increase their delivery. I recommend it for publication after the following minor points are fixed.

Authors’ response to general comments: We are very thankful for the positive appreciation of our draft. We would like to thank the referee for the constructive comments on our manuscript. We carefully examined the comments and revised the manuscript, according to the referees’ suggestions.

Point 1: Drug loading content for different drugs should be calculated.

Response 1: The authors are very thankful for the suggestion. The DL (drug loading) was calculated taking into account the total dry weight of PLGA NPs using the following equation: DL (%) = [(Total amount of DCA – Free DCA in supernatant) / Total dry weight of NPs] x 100.

This text and DL formula are now part of the manuscript (page 6, line 417 and page 7, lines 420-422).

Point 2: The authors should add explanations about the different roles of PLGA-b-poly(ethylene glycol)-maleimide and Pluronic® F127 during nanoparticle preparation.

Response 2: We thank the referee’s suggestion. PLGA is one of the best characterized biodegradable and biocompatible copolymer that breaks down into non-toxic products (H2O and CO2) that are eliminated from the body. Surface modification with PEG (PEGylation) increases the formulation hydrophilicity, as well as physiological stability against undesired aggregation and premature elimination. Pluronic® F127 is a surfactant polyol used to further stabilize the colloidal dispersion of PLGA NPs, and adjust the formulation parameters regarding desired size range.

These explanations about the different roles of PLGA-PEG (page 6, lines 395-399) and Pluronic® F127 (page 6, lines 402-404) are now part of the manuscript (page 6).

Point 3: Why were the zeta potential of nanoparticles negative?

Response 3: The negative charge of PLGA NP is due to its deprotonated carboxylic end groups of lactic and glycolic acid. Still, the slight negative values founded are close to the neutrality, in agreement with the PEGuilation effect on the stealth character of PEGuilated nanoparticles. The negative value in the range of -4 to -8 mV is not biologically relevant.

Point 4: In this study, targeting groups were not modified on the nanoparticles, why was PLGA-b-poly(ethylene glycol)-maleimide used but not PLGA-b-poly(ethylene glycol)?

Response 4: We are very thankful for this remark. In fact, we only used PLGA-b-poly(ethylene glycol), but by error it was written PLGA-b-poly(ethylene glycol)-maleimide. This error has already been corrected in the manuscript (page 6, line 394).

Point 5: In the introduction, 'Among polymeric NPs', other polymeric NPs (Langmuir 35 (5), 1273-1283; Colloids and Surfaces B: Biointerfaces, 189, 110830; Pharmaceutics 13.8 (2021): 1319) are encouraged to be included.

Response 5: We thank the recommendation and we added the others polymeric NPs and the references indicated (page 3, lines 200,201).